# Generative Adversarial Network (Generative Artificial Intelligence) in Pediatric Radiology: A Systematic Review

**DOI:** 10.3390/children10081372

**Published:** 2023-08-10

**Authors:** Curtise K. C. Ng

**Affiliations:** 1Curtin Medical School, Curtin University, GPO Box U1987, Perth, WA 6845, Australia; curtise.ng@curtin.edu.au or curtise_ng@yahoo.com.hk; Tel.: +61-8-9266-7314; Fax: +61-8-9266-2377; 2Curtin Health Innovation Research Institute (CHIRI), Faculty of Health Sciences, Curtin University, GPO Box U1987, Perth, WA 6845, Australia

**Keywords:** computer-aided diagnosis, data augmentation, deep learning, dose reduction, image reconstruction, image segmentation, image translation, machine learning, medical imaging, noise

## Abstract

Generative artificial intelligence, especially with regard to the generative adversarial network (GAN), is an important research area in radiology as evidenced by a number of literature reviews on the role of GAN in radiology published in the last few years. However, no review article about GAN in pediatric radiology has been published yet. The purpose of this paper is to systematically review applications of GAN in pediatric radiology, their performances, and methods for their performance evaluation. Electronic databases were used for a literature search on 6 April 2023. Thirty-seven papers met the selection criteria and were included. This review reveals that the GAN can be applied to magnetic resonance imaging, X-ray, computed tomography, ultrasound and positron emission tomography for image translation, segmentation, reconstruction, quality assessment, synthesis and data augmentation, and disease diagnosis. About 80% of the included studies compared their GAN model performances with those of other approaches and indicated that their GAN models outperformed the others by 0.1–158.6%. However, these study findings should be used with caution because of a number of methodological weaknesses. For future GAN studies, more robust methods will be essential for addressing these issues. Otherwise, this would affect the clinical adoption of the GAN-based applications in pediatric radiology and the potential advantages of GAN could not be realized widely.

## 1. Introduction

Artificial intelligence (AI) is a popular topic in radiology such as for rapid disease (e.g., COVID-19) detection on various platforms including mobile devices [1,2,3,4,5,6,7,8,9,10,11,12]. Additionally, the number of AI research articles in radiology has grown exponentially over recent years [1,2]. Various commercial AI products have been available for applications in clinical practice such as radiological examination dose optimization [13,14,15,16,17,18,19,20,21,22,23,24,25,26], computer-aided detection and diagnosis (CAD) [27,28,29,30,31,32,33,34,35,36,37,38,39,40,41,42,43,44,45,46,47,48], and medical image segmentation [49,50,51,52,53]. Predominantly, these applications in radiology are based on deductive AI techniques [1,13,14,15,16,17,18,19,20,21,22,23,24,25,26,27,28,29,30,31,32,33,34,35,36,37,38,39,40,41,42,43,44,45,46,47,48,49,50,51,52,53,54]. However, generative AI, especially the generative adversarial network (GAN) which focuses on the creation of new and original content, has started attracting the attention of radiology researchers and clinicians as evidenced by a number of literature reviews on the role of GAN in radiology published in the last few years [54,55,56,57,58,59,60,61,62].

The GAN was devised by Goodfellow et al. in 2014 [56,59,62,63]. Its basic form (also known as Vanilla GAN) consists of two models, a generator and a discriminator. Development of this GAN model requires training the generator to produce fake images while the discriminator is responsible for determining whether the image produced by the generator is fake or real. The training is completed upon the discriminator unable to indicate the generator’s output images are fake, and hence the generator becomes capable of producing high-quality fake images close to the real ones [56,59,62,63,64,65]. This capability is highly relevant to medical imaging and therefore radiology [64,65]. Its current applications in radiology include image synthesis and data augmentation [1,55,56,57,59,60,61,62], image translation (e.g., from one modality to another one [1,55,56,58,59,60,61,62], from normal to abnormal [1,55,62], etc.), image reconstruction (e.g., denoising [1,55,59,60,61], artifact removal [1,56,58,61], super-resolution (image spatial resolution improvement) [1,55,56,57,59,61,64,65], motion unsharpness correction [61], etc.), image feature extraction [55,57,60,61], image segmentation [1,55,56,57,60,61,62], anomaly detection [55,56,60], disease diagnosis [55,57,60], prediction [55,56,61] and prognosis [55,57,60,61], and image registration [1,55,60,61].

Pediatric radiology is a subset of radiology [26,28,29,66,67]. The aforementioned review findings may not be applicable to pediatric radiology [28,29,55,56,57,58,59,60,61,62,67]. For example, the application of GAN for prostate cancer segmentation appears not relevant to children [60,68]. Although several literature reviews about AI in pediatric radiology have been published, none of them focused on the GAN [26,28,29,67]. Given that the GAN is an important topic area in radiology and the recent literature reviews focused on its applications in this discipline, it is timely to conduct a systematic review of its applications in pediatric radiology [29,55,56,57,58,59,60,61,62]. The purpose of this article is to systematically review published original studies to answer the question “What are the applications of GAN in pediatric radiology, their performances, and methods for their performance evaluation?”.

## 2. Materials and Methods

This systematic review of the GAN in pediatric radiology was carried out according to the PRISMA guidelines and patient/population, intervention, comparison, and outcome (PICO) model (Table 1) [26,29,69]. Four major processes, literature search, article selection, and data extraction and synthesis were involved [26,29].

### 2.1. Literature Search

The electronic scholarly publication databases, *EBSCOhost/Cumulative Index of Nursing and Allied Health Literature (CINAHL) Ultimate*, *Ovid/Embase*, *PubMed/Medline*, *ScienceDirect*, *Scopus*, *SpringerLink*, *Web of Science*, and *Wiley Online Library* were used for literature search on 6 April 2023 to identify articles about the GAN in pediatric radiology and publication year was not restricted. The search statement, (“Generative Adversarial Network” OR “Generative Artificial Intelligence”) AND (“Pediatric” OR “Children”) AND (“Radiology” OR “Medical Imaging”) was used. The review focus was used to derive the search keywords [26,29].

### 2.2. Article Selection

Article selection was conducted by one reviewer with a literature review experience of more than 20 years [26,29,70]. Table 2 shows the article’s inclusion and exclusion criteria.

The exclusion criteria of Table 2 were established because of: 1. unavailability of well-developed methodological guidelines for appropriate grey literature selection; 2. Incomplete study information given in conference abstracts; 3. a lack of primary evidence in editorials, reviews, perspectives, opinions, and commentary; and 4. unsubstantiated information given in non-peer-reviewed papers [26,29,62,71]. The detailed process of the article selection is shown in Figure 1 [26,29,69]. Duplicate papers were first removed from the database search results. Subsequently, article titles, abstracts, and full texts were assessed against the selection criteria. Each non-duplicate paper in the search results was kept unless a decision on its exclusion could be made. Additionally, relevant articles were identified by checking reference lists of the included papers [26,29,71].

### 2.3. Data Extraction and Synthesis

Three systematic reviews on the GAN for image classification and segmentation in radiology [62], AI for radiation dose optimization [26] and CAD in pediatric radiology [29], and one narrative review about the GAN in adult brain imaging [56] were used to develop a data extraction form (Table 3). The data, author name and country, publication year, imaging modality, GAN architecture (such as cycle-consistent GAN (CycleGAN)), study design (either prospective or retrospective), patient/population (e.g., 0–10-year-old children), dataset source (such as public cardiac magnetic resonance imaging (MRI) dataset by Children’s Hospital Los Angeles, USA) and size (e.g., total: 33 scans-training: 25; validation: 4; testing: 4, etc.), any sample size calculation, application area (such as image synthesis and data augmentation), model commercial availability, model internal validation type (e.g., 4-fold cross-validation, etc.), any model external validation (i.e., any testing of model based on dataset not used in internal validation and obtained from different setting), reference standard for establishing ground truth (such as expert consensus), any comparison of performance of model with clinician, and key findings of model performance (e.g., area under receiver operating characteristic curve (AUC), sensitivity, specificity, positive predictive value (PPV), negative predictive value (NPV), accuracy, and F1 score, etc.) were extracted from every article included [26,29,56,62]. For facilitating GAN model performance comparison, improvement figures such as improvement percentages when the GAN was used were synthesized (if not reported) based on the available absolute figures (if feasible) [26]. When a study reported performances for more than one GAN model, only the best-performing model performance values were shown [29,72]. Meta-analysis was not performed as this systematic review included a range of GAN applications, resulting in high study heterogeneity which would affect its usefulness [29,73,74,75]. The quality assessment tool for studies with diverse designs (QATSDD) was used to determine quality percentages for all included papers [26,71,76]. <50%, 50–70%, and >70% represented low, moderate, and high qualities of study, respectively [26,71].

## 3. Results

Thirty-seven papers that met the selection criteria were included in this review. These study characteristics are shown in Table 3. All identified articles were published over the last five years and the publication number increased every year with the highest number in 2022 [77,78,79,80,81,82,83,84,85,86,87,88,89,90,91,92,93,94,95,96,97,98,99,100,101,102,103,104,105,106,107,108,109,110,111,112,113]. This increasing trend was in line with the one in radiology [1,77,78,79,80,81,82,83,84,85,86,87,88,89,90,91,92,93,94,95,96,97,98,99,100,101,102,103,104,105,106,107,108,109,110,111,112,113]. About half of the articles (*n* = 17) were journal papers [77,78,82,84,87,90,92,97,98,99,100,101,102,103,105,109,111]. Around two-thirds of these (*n* = 11) were determined as being of high quality [82,84,87,90,92,97,102,103,105,109,111]. All low-quality ones were conference papers (*n* = 12) [79,80,81,83,85,86,91,93,94,95,104,108]. The GAN was commonly applied to MRI (*n* = 18) [77,78,83,84,87,90,97,101,103,104,105,106,108,109,110,111,112,113] and X-ray (*n* = 13) [79,80,89,91,92,94,95,96,98,99,100,102,107], and the others included computed tomography (CT) (*n* = 4) [82,86,93,97], ultrasound (*n* = 2) [85,88] and positron emission tomography (PET) (*n* = 1) [81]. Although the basic GAN architecture was still popular among the included studies (*n* = 11) [77,78,80,82,83,84,89,94,97,99,106], its variant, cycle-consistent GAN (CycleGAN), was the second most common (*n* = 10) [101,102,103,104,107,108,109,110,111,112].

**Table 3 children-10-01372-t003:** Characteristics of generative adversarial network (GAN) studies in pediatric radiology (grouped by their applications).

Author, Year & Country	Modality	GAN Architecture	Study Design	Patient/Population	Dataset Source	Dataset Size	Sample Size Calculation	Application Area	Commercial Availability	Internal Validation Type	External Validation	Reference Standard	AI VS Clinician	Key Findings
Disease Diagnosis														
Kuttala et al. (2022)—Australia, India, and the United Arab Emirates [77]	MRI	GAN	Retrospective	Children (median ages: 12.6 (baseline) and 15.0 (follow-up) years	Public brain MRI dataset (Autism Brain Imaging Data Exchange II)	Total: 70 scans-training: 24; testing: 46	No	Autism diagnosis based on brain MRI images	No	NR	No	NR	No	158.6% accuracy (U-Net: 0.370; GAN: 0.957) and 114.3% AUC (U-Net: 0.420; GAN: 0.900) improvements for autism diagnoses, respectively
Kuttala et al. (2022)—Australia, India, and the United Arab Emirates [78]	MRI	GAN	Retrospective	Children (median ages: 12 (baseline) and 15 (follow-up) years	Public brain MRI datasets (ADHD-200 and Autism Brain Imaging Data Exchange II)	Total: 265 scans-training: 48; testing: 217	No	ADHA and autism diagnosis based on brain MRI images	No	NR	No	NR	No	29.6% and 39.7% accuracy improvements for ADHD and autism diagnoses (3D CNN: 0.659 and 0.700; GAN: 0.854 and 0.978), respectively. GAN AUC: 0.850 (ADHD) and 0.910 (autism)
Motamed and Khalvati (2021)—Canada [79]	X-ray	DCGAN	Retrospective	1–5-year-old children	Public CXR dataset by Guangzhou Women and Children’s Medical Center, China	Total: 4875 images-training: 3875; testing: 1000	No	Pneumonia diagnosis based on CXR	No	NR	No	NR	No	3.5% AUC improvement (Deep SVDD: 0.86; DCGAN: 0.89)
Image Reconstruction														
Dittimi and Suen (2020)—Canada [80]	X-ray	GAN	Retrospective	1–5-year-old children	Public CXR dataset by Guangzhou Women and Children’s Medical Center, China	Total: 5863 images	No	CXR image reconstruction (super-resolution)	No	70:30 random split	No	Original CXR images	No	19.1% SSIM (SRCNN: 0.832; SRCNN-GAN: 0.991) and 46.5% PSNR (SRCNN: 26.18; SRCNN-GAN: 38.36 dB) improvements
Fu et al. (2022)—China [81]	PET	TransGAN	Retrospective	Children	Private brain PET dataset by Hangzhou Universal Medical Imaging Diagnostic Center, China	Total: 45 scans	No	Brain PET image reconstruction (denoising)	No	10-fold cross-validation	No	Original full-dose PET images	No	10.3% SSIM (U-Net: 0.861; TransGAN-SDAM: 0.950) and 29.9% PSNR (U-Net: 26.1; TransGAN-SDAM: 33.9 dB) improvements with 67.7% VSMD reduction (U-Net: 0.133; TransGAN-SDAM: 0.043)
Park et al. (2022)—Republic of Korea [82]	CT	GAN	Retrospective	3 groups of children (mean ages (years): 6.2 ± 2.2; 7.2 ± 2.5; 7.4 ± 2.2)	Private abdominal CT dataset	Total: 3160 images-training: 1680; validation: 820; testing: 660	No	Low-dose abdominal CT image reconstruction (denoising)	No	NR	Yes	Consensus of 1 pediatric and 1 abdominal radiologist (6 and 8 years’ experiences), respectively.	Yes	42.7% noise reduction (LDCT: 12.4 ± 5.0; SAFIRE: 9.5 ± 4.0; GAN: 7.1 ± 2.7), and 39.3% (portal vein) and 45.8% (liver) SNR (LDCT: 22.9 ± 9.3 and 13.1 ± 5.7; SAFIRE: 30.1 ± 12.2 and 17.3 ± 7.6; GAN: 31.9 ± 13.0 and 19.1 ± 7.9) and 30.9% (portal vein) and 32.8% (liver) CNR (LDCT: 16.2 ± 7.5 and 6.4 ± 3.7; SAFIRE: 21.2 ± 9.8 and 8.5 ± 5.0; GAN: 21.2 ± 10.1 and 8.5 ± 4.3) improvements when compared with LDCT images, respectively.
Pham et al. (2019)—France [83]	MRI	3D GAN	Retrospective	Neonates	Public (Developing Human Connectome Project) and private brain MRI datasets by Reims Hospital, France	Total: 40 images-training: 30; testing: 10	No	Brain MRI image reconstruction (super-resolution) and segmentation	No	NR	Yes	NR	No	1.39% SSIM (non-DL: 0.9492; SRCNN: 0.9739; GAN: 0.9624) and 3.42% PSNR (non-DL: 30.70 dB; SRCNN: 35.84 dB; GAN: 31.75 dB) improvements for super-resolution and 12.4% DSC improvement for segmentation (atlas-based: 0.788; intensity-based: 0.818; GAN: 0.886) when compared with non-DL approaches, respectively
Image Segmentation														
Decourt and Duong (2020)—Canada and France [84]	MRI	GAN	Retrospective	2–18-year-old children	Private cardiac MRI dataset by Hospital for Sick Children in Toronto, Canada	Total: 33 scans-training: 25; validation: 4; testing: 4	No	Cardiac MRI image segmentation	No	Cross-validation	Yes	Manual segmentation by clinicians	No	2.4% mean DSC improvement (U-Net: 0.85; GAN: 0.87) with 3.8% mean HD reduction (U-Net: 2.55 mm; GAN: 2.46 mm)
Guo et al. (2019)—China [85]	US	DNGAN	NR	0–10-year-old children	Private echocardiography dataset by a Chinese hospital	Total: 87 scans-training: 1765 images; testing: 451 images	No	Echocardiography image segmentation	No	NR	No	NR	No	4.6% mean DSC (U-Net: 0.88; DNGAN: 0.92), 7.6% mean Jaccard index (U-Net: 0.80; DNGAN: 0.86) and 8.5% mean PPV (U-Net: 0.86; DNGAN: 0.94) improvements but with 0.9% mean sensitivity reduction (U-Net: 0.93; DNGAN: 0.92)
Kan et al. (2021)—USA [86]	CT	AC-GAN	NR	1–17-year-old children	Private abdominal CT dataset by Medical College of Wisconsin, USA	Total: 64 scans	No	Abdominal CT image segmentation	No	4-fold cross-validation	No	NR	No	3.9% and 0.7% mean DSC improvements (U-Net: 0.697 and 0.923; GAN: 0.724 and 0.929) with 35.0% and 13.3% mean HD reductions (U-Net: 1.090 and 0.390 mm; GAN: 0.709 and 0.338 mm) for uterus and prostate segmentations, respectively
Karimi-Bidhendi et al. (2020)—USA [87]	MRI	DCGAN	Retrospective	2–18-year-old children	Public cardiac MRI datasets by Children’s Hospital Los Angeles, USA, and ACDC	Total: 159 scans-training: 41; testing: 118	No	Cardiac MRI image segmentation	No	80:20 random split	Yes	Manual image segmentation by a pediatric cardiologist sub-specialized in cardiac MRI	No	34.5% mean DSC (cvi42: 0.631; U-Net: 0.782; DCGAN: 0.848), 38.5% Jaccard index (cvi42: 0.556; U-Net: 0.702; DCGAN: 0.770), 53.2% R^2^ (cvi42: 0.629; U-Net: 0.871; DCGAN: 0.963), 30.8% sensitivity (cvi42: 0.666; U-Net: 0.775; DCGAN: 0.872), 0.1% specificity (cvi42: 0.997; U-Net: 0.998; DCGAN: 0.998), 34.0% PPV (cvi42: 0.636; U-Net: 0.839; DCGAN: 0.852) and 0.4% NPV (cvi42: 0.995; U-Net: 0.997; DCGAN: 0.998) improvements with 24.7% mean HD (cvi42: 11.0 mm; U-Net: 11.0 mm; DCGAN: 8.3 mm) and 31.6% MCD reductions (cvi42: 4.4 mm; U-Net: 4.5 mm; DCGAN: 3.0 mm) when compared with cvi42
Zhou et al. (2022)—Canada [88]	US	pix2pix GAN	Prospective	Children	Private wrist US dataset by University of Alberta Hospital, Canada	Total: 57 scans-training: 47; testing: 10	No	Wrist US image segmentation	No	NR	No	Manual segmentation by radiologist and sonographer with 18 and 7 years’ experience, respectively	No	7.5% sensitivity improvement (U-Net: 0.642; GAN: 0.690) but with 5.6% DSC (U-Net: 0.698; GAN: 0.659), 8.6% Jaccard index (U-Net: 0.548; GAN: 0.501) and 17.8% PPV (U-Net: 0.783; GAN: 0.644) reductions
Image Synthesis and Data Augmentation														
Banerjee et al. (2021)—India [89]	X-ray	GAN	Retrospective	1–5-year-old children	Public CXR dataset by Guangzhou Women and Children’s Medical Center, China	Total: 5863 images	No	CXR image synthesis and data augmentation for DL-CAD model training	No	NR	No	NR	No	13,921 images were generated for training the DL-CAD model for pneumonia with 6.3% accuracy improvement (with and without GAN: 0.986 and 0.928), respectively
Diller et al. (2020)—Germany [90]	MRI	PG-GAN	Retrospective	Children with a median age of 15 years (IQR: 12.8–19.3 years)	Private cardiac MRI dataset by German Competence Network for Congenital Heart Defects	Total: 303 scans	No	Cardiac MRI image synthesis and data augmentation	No	NR	No	Ground truth determined by researchers	Yes	Mean rates of PG-GAN generated images identified by clinicians being fake: 70.5% (3 cardiologists) and 86.7% (2 cardiac MRI experts)
Guo et al. (2021)—China [91]	X-ray	AC-GAN	Retrospective	1–5-year-old children	Public CXR dataset by Guangzhou Women and Children’s Medical Center, China	Total: 5856 images-training: 1500; testing: 4356	No	CXR image synthesis and data augmentation for DL-CAD model training	No	NR	No	NR	No	250 pneumonia and 250 normal images generated for DL-CAD model training with 0.6% accuracy improvement (with and without AC-GAN: 0.913 and 0.907), respectively
Guo et al. (2022)—China [92]	X-ray	AC-GAN	Prospective	2–14-year-old children	Private CXR dataset by Quanzhou Women’s and Children’s Hospital, China	Total: 6442 images-training: 3600	No	CXR image synthesis and data augmentation for DL-CAD model training	No	NR	No	NR	No	2000 images generated with 7.7% and 13.5% differences between ground truth (IS: 2.08) and AC-GAN generated normal (IS: 1.92) and pneumonia (IS: 1.80) images, respectively. The use of AC-GAN images for training the DL-CAD model improved sensitivity (with and without AC-GAN: 0.86 and 0.62), specificity (with and without AC-GAN: 0.97 and 0.90), and accuracy (with and without AC-GAN: 0.91 and 0.76) by 38.7%, 7.8%, and 19.7%, respectively
Kan et al. (2020)-USA [93]	CT	AC-GAN	NR	1–18-year-old children	NR	Total: 5 scans	No	Pancreatic CT image synthesis and data augmentation	No	NR	No	NR	No	AC-GAN was able to generate high-resolution pancreas images with fine details and without any streak artifact and irregular pancreas contour when compared with DCGAN
Khalifa et al. (2022)-Egypt [94]	X-ray	GAN	Retrospective	1–5-year-old children	Public CXR dataset by Guangzhou Women and Children’s Medical Center, China	Total: 624 images	No	CXR image synthesis and data augmentation for DL-CAD model training	No	80:20 random split	No	Specialist consensus	No	5616 images generated for training the DL-CAD model for pneumonia with 6.7% accuracy improvement (with and without GAN: 0.990 and 0.928), respectively
Kora Venu (2021)-USA [95]	X-ray	DCGAN	Retrospective	1–5 years old children	Public CXR dataset by Guangzhou Women and Children’s Medical Center, China	Total: 5856 images-training: 4684; testing: 1172	No	CXR image synthesis and data augmentation for DL-CAD model training	No	80:20 random split	No	NR	No	2152 images generated for training DL-CAD model for pneumonia with 2.6% AUC (with and without DCGAN: 0.993 and 0.968), 6.5% sensitivity (with and without DCGAN: 0.993 and 0.932), 13.5% PPV (with and without DCGAN: 0.990 and 0.872), 6.4% accuracy (with and without DCGAN: 0.987 and 0.928) and 10.0% F1 score improvements (with and without DCGAN: 0.991 and 0.901), respectively
Li and Ke (2022)-USA [96]	X-ray	DCGAN	Retrospective	1–5 years old children	Public CXR dataset by Guangzhou Women and Children’s Medical Center, China	Total: 5910 images-training: 4300; validation: 724; testing: 886	No	CXR image synthesis and data augmentation for DL-CAD model training	No	90:10 random split	No	NR	No	2700 images generated for training DL-CAD model for pneumonia with 13.7% accuracy (with and without DCGAN: 0.960 and 0.844) and 1.1% AUC (with and without DCGAN: 0.994 and 0.983) improvements, respectively
Prince et al. (2020)-Canada and USA [97]	CT and MRI	GAN	Retrospective	Children	Public (ATPC Consortium) and private brain CT-MRI datasets by Children’s Hospital Colorado and St. Jude Children’s Research Hospital, USA	Total: 86 CT-MRI scans-training: 53; testing: 33	No	Brain CT-MRI image synthesis and data augmentation for DL-CAD model training	No	60:40 random split and 5-fold cross-validation	No	Histology	Yes	2000 CT and 2000 MRI images generated for training DL-CAD model for adamantinomatous craniopharyngioma with 0.890 (CT) and 0.974 (MRI) accuracy. 17.0% AUC improvement for MRI (radiologists: 0.833; GAN: 0.975) but 1.6% AUC reduction for CT (radiologists: 0.894; GAN: 0.880).
Su et al. (2021)-China [98]	X-ray	WGAN	Retrospective	1–19 years old children	Public hand X-ray dataset (RSNA Pediatric Bone Age Challenge)	Total: 14,236 images-training: 12,611; validation: 1425; testing: 200	No	Hand X-ray image synthesis and data augmentation, and bone age assessment	No	NR	No	Manual assessment by expert clinicians	No	11,350 images generated with 7.9 IS, 17.3 FID and 20.0% MAE reduction (CNN: 5.29 months; WGAN: 4.23 months)
Szepesi and Szilágyi (2022)-Hungary and Romania [99]	X-ray	GAN	Retrospective	1–5 years old children	Public CXR dataset by Guangzhou Women and Children’s Medical Center, China	Total: 5856 images-training: 4099; validation: 586; testing: 1171	No	CXR image synthesis and data augmentation for DL-CAD model training	No	10-fold cross-validation	No	Expert clinicians	No	2152 images generated for training DL-CAD model for pneumonia with 0.9820 AUC, 0.9734 sensitivity, 0.9740 PPV, 0.9721 accuracy, and 3.9% F1 score improvement (CNN: 0.9375; GAN: 0.9740)
Vetrimani et al. (2023)-India [100]	X-ray	DCGAN	Retrospective	1–8 years old children	Public CXR datasets by Guangzhou Women and Children’s Medical Center, China and from various websites such as Radiopaedia	Total: 987 images-training: 645; validation: 342	No	CXR image synthesis and data augmentation for DL-CAD model training	No	NR	No	NR	No	Additional images generated by DCGAN for training DL-CAD model for laryngotracheobronchitis with 0.8791 sensitivity, 0.854 PPV, 0.8832 accuracy and 0.8666 F1 score.
Image Translation														
Chen et al. (2021)-China and USA [101]	MRI	3D CycleGAN	Retrospective	Neonates	Private brain MRI datasets by Xi’an Jiaotong University, China and University of North Carolina, USA	Total: 40 images	No	Image translation (for domain adaptation in brain MRI image segmentation)	No	NR	No	NR	No	1.2% mean DSC improvement (with and without 3D CycleGAN: 0.86 and 0.85) with 12.8% mean HD (with and without 3D CycleGAN: 13.03 and 14.94 mm) and 16.0% MSD (with and without 3D CycleGAN: 0.23 and 0.27 mm) reductions, respectively
Hržić et al. (2021)-Austria, Croatia and Germany [102]	X-ray	CycleGAN	Retrospective	Children (mean age: 11 ± 4 years)	Private wrist X-ray dataset by Medical University of Graz, Austria	Total: 9672 images- training: 7600; validation: 636; testing: 1436	No	Wrist X-ray image translation (cast suppression)	No	NR	No	Real castless wrist X-ray images	No	Real castless and CycleGAN generated cast suppressed image histogram similarity scores: 0.998 (correlation) and 222,503 (intersection) with difference values: 59,451 (chi-square distance) and 0.147 (Hellinger distance)
Kaplan et al. (2022)-USA and Germany [103]	MRI	3D CycleGAN	Prospective	Neonates (mean PMA: 41.1 ± 1.5 weeks) and infants (mean age: 41.2 ± 1.9 weeks)	Private brain MRI datasets by Washington University and ECHO Program, USA	Total: 137 scans-training: 107; testing: 30	No	Brain MRI image translation (T1w-to-T2w)	No	NR	Yes	Real T2w MRI images acquired from same patients	No	9.7% and 7.9% SSIM and DSC improvements (Kaplan-T2w: 0.72 and 0.76; CycleGAN: 0.79 and 0.82) with 18.8% relative MAE reduction (Kaplan-T2w: 6.9; CycleGAN: 5.6) and no statistically significant CNR difference (Kaplan-T2w: 0.76; CycleGAN: 0.63; original images: 0.62), respectively
Khalili et al. (2019)-The Netherlands [104]	MRI	CycleGAN	NR	Neonates (mean PMA: 30.7 ± 1.0 weeks)	Private brain MRI dataset by University Medical Center Utrecht, The Netherlands	Total: 80 scans-training: 35; testing: 45	No	Brain MRI image translation between motion blurred and blurless ones for training DL-segmentation model	No	NR	No	NR	No	6.7% DSC improvement (with and without CycleGAN: 0.80 and 0.75) with 32.4% HD (with and without CycleGAN: 25.0 and 37.0 mm) and 60.5% MSD reductions (with and without CycleGAN: 0.5 and 1.3 mm) for segmentation, respectively. Median subjective image quality and segmentation accuracy ratings (scale 1–5): before (2 and 3) and after motion unsharpness correction (3 and 4), respectively
Maspero et al. (2020)-The Netherlands [105]	MRI	2D CGAN	Retrospective	2.6–19 (mean: 10 ± 5) years old children	Private brain CT and T1w MRI dataset by University Medical Center Utrecht, The Netherlands	Total: 60 CT and MRI scans-training: 30; validation: 10; testing: 20	No	Brain MRI image translation to CT for radiation therapy planning	No	4-fold cross-validation	No	Real CT images acquired from same patients	No	DSC: 0.92; MAE: 61 HU for CT images generated from MRI images by CGAN
Peng et al. (2020)-China, Japan and USA [106]	MRI	3D GAN	Retrospective	6–12 months old children	Public brain MRI dataset (Infant Brain Imaging Study)	Total: 578 scans-training: 462; validation: 58; testing: 58	No	Brain MRI image translation between images acquired 6 months apart	No	NR	No	Real MRI images acquired from same patient 6 months apart	No	1.5% DSC improvement (U-Net: 0.809; GAN: 0.821) and 7.5% MSD reduction (U-Net: 0.577 mm; GAN: 0.534 mm) but with 16.8% RVD increase (U-Net: 0.0424; GAN: 0.0495)
Tang et al. (2019)-China and USA [107]	X-ray	CycleGAN	Retrospective	1–5 years old children and adult	Public CXR datasets by Guangzhou Women and Children’s Medical Center, China and from RSNA Pneumonia Detection Challenge	Total: 17,508 images-training: 16,884; testing: 624	No	Image translation (for domain adaptation of DL-CAD)	No	5-fold cross-validation	No	NR	No	7.8% AUC (with and without CycleGAN: 0.963 and 0.893), 11.1% sensitivity (with and without CycleGAN: 0.929 and 0.836), 12.7% specificity (with and without CycleGAN: 0.911 and 0.808), 12.8% accuracy (with and without CycleGAN: 0.931 and 0.825) and 8.1% F1 score (with and without CycleGAN: 0.930 and 0.860) improvements, respectively
Tor-Diez et al. (2020)-USA [108]	MRI	CycleGAN	NR	Children	Private brain MRI datasets by Children’s National Hospital, Children’s Hospital of Philadelphia and Children’s Hospital of Colorado, USA	Total: 18 scans	No	Image translation (for domain adaptation in brain MRI image segmentation)	No	Leave-one-out cross-validation	No	NR	No	18.3% DSC improvement for anterior visual pathway segmentation (U-Net: 0.509; CycleGAN: 0.602)
Wang et al. (2021)-USA [109]	MRI	CycleGAN	Retrospective	2 groups of children (median ages: 8.3 and 6.4 years; ranges: 1–20 and 2–14 years), respectively	Private brain CT and T1w MRI datasets by St Jude Children’s Research Hospital, USA	Total: 132 CT and MRI scans-training: 125; testing: 7	No	Brain MRI image translation to CT for radiation therapy planning	No	NR	No	Real CT images acquired from same patients	No	SSIM: 0.90; DSC of air/bone: 0.86/0.81; MAE: 65.3 HU; PSNR: 28.5 dB for CT images generated from MRI images by CycleGAN
Wang et al. (2021)-USA [110]	MRI	CycleGAN	Retrospective	1.1–21.3 years old children and adult	Private brain and pelvic CT and MRI datasets by St Jude Children’s Research Hospital, USA	Total: 141 CT and MRI scans; training: 136; testing: 5	No	Pelvic MRI image translation to CT for radiation therapy planning	No	NR	No	Real CT images acquired from same patients	No	Mean SSIM: 0.93 and 0.93; MAE: 52.4 and 85.4 HU; ME: −3.4 and −6.6 HU; PSNR: 30.6 and 29.2 dB for CT images generated from T1w and T2w MRI images by CycleGAN, respectively
Wang et al. (2022)-USA [111]	MRI	CycleGAN	Retrospective	1.1–20.3 (median: 9.0) years old children and adult	Private brain CT and MRI datasets by St. Jude Children’s Research Hospital, USA	Total: 195 CT and MRI scans-training: 150; testing: 45	No	Brain MRI image translation to CT and RPSP images for radiation therapy planning	No	NR	No	Real CT images acquired from same patients	No	SSIM: 0.92 and 0.91; DSC of air/bone: 0.98/0.83 and 0.97/0.85 MAE: 44.1 and 42.4 HU; ME: 8.6 and 18.8 HU; PSNR: 32.6 and 31.5 dB for CT images generated from T1w and T2w MRI images by CycleGAN, respectively
Zhao et al. (2019)-China and USA [112]	MRI	CycleGAN	Retrospective	0–2 years old children	Public brain MRI dataset (UNC/UMN Baby Connectome Project)	Total: 360 scans-training: 252; testing: 108	No	Image translation (for domain adaptation)	No	NR	No	Original MRI images	No	14.1% PSNR improvement (non-DL: 29.00 dB; CycleGAN: 33.09 dB) and 33.9% MAE reduction (non-DL: 0.124; CycleGAN: 0.082) for domain adaptation
Other														
Mostapha et al. (2019)-USA [113]	MRI	3D DCGAN	Retrospective	1–6-year-old children	Public brain MRI datasets (UNC/UMN Baby Connectome Project and UNC Early Brain Development Study)	Total: 2187 scans	No	Automatic brain MRI image quality assessment	No	80:20 random split	No	Manual image quality assessment by MRI experts	No	92.9% sensitivity (VAE: 0.42; DCGAN: 0.81), 2.2% specificity (VAE: 0.93; DCGAN: 0.95), and 47.6% accuracy (VAE: 0.63; DCGAN: 0.93) improvements for automatic image quality assessment, respectively

2D, two-dimensional; 3D, three-dimensional; AC-GAN, auxiliary classifier generative adversarial network; ACDC, Automated Cardiac Diagnosis Challenge of 2017 Medical Image Computing and Computer Assisted Intervention; ADHD, attention deficit hyperactivity disorder; AI, artificial intelligence; AIGAN, attention-encoding integrated generative adversarial network; ATPC, Advancing Treatment for Pediatric Craniopharyngioma; AUC, area under the receiver operating characteristic curve; CAD, computer-aided detection and diagnosis; CGAN, conditional generative adversarial network; CNN, convolutional neural network; CNR, contrast-to-noise ratio; cvi42, a commercial deep learning-based segmentation product (Circle Cardiovascular Imaging, Calgary, Alberta, Canada); CT, computed tomography; CXR, chest X-ray; CycleGAN, cycle-consistent generative adversarial network; DCGAN, deep convolutional generative adversarial network; DL, deep learning; DNGAN, dual network generative adversarial network; DSC, Dice similarity coefficient; ECHO, Environmental Influences on Child Health Outcomes; FID, Fréchet inception distance; HD, Hausdorff distance; HU, Hounsfield unit; IQR, interquartile range; IS, inception score; Kaplan-T2w, a registration-based method for T1w-to-T2w translation; LDCT, low-dose computed tomography; MAE, mean absolute error; MCD, mean contour distance; ME, voxel-based mean error; MRI, magnetic resonance imaging; MSD, mean surface distance; NPV, negative predictive value; NR, not reported; PET, positron emission tomography; PG-GAN, progressive generative adversarial network; PMA, postmenstrual age; PPV, positive predictive value; PSNR, peak signal to noise ratio; R^2^, coefficient of determination; RPSP, relative proton stopping power; RSNA, Radiological Society of North America; RVD, relative volume difference; SAFIRE, sinogram affirmed iterative reconstruction; SDAM, spatial deformable aggregation module; SNR, signal-to-noise ratio; SRCNN, super-resolution convolutional neural network; SSIM, structural index similarity; SVDD, support vector data description; T1w, T1-weighted; T2w, T2-weighted; TransGAN, transformer-based generative adversarial network; UMN, University of Minnesota; UNC, University of North Carolina; US, ultrasound; VAE, variational autoencoder; VSMD, voxel-scale metabolic difference; WGAN, Wasserstein generative adversarial network.

Both image synthesis and data augmentation (*n* = 12) [89,90,91,92,93,94,95,96,97,98,99,100], and image translation (*n* = 12) [101,102,103,104,105,106,107,108,109,110,111,112] were the commonest application areas of GAN in pediatric radiology. Other GAN application areas included image segmentation (*n* = 5) [84,85,86,87,88], image reconstruction (*n* = 4) [80,81,82,83], disease diagnosis (*n* = 3) [77,78,79], and image quality assessment (*n* = 1) [113]. However, none of the GAN models involved in these studies were commercially available [77,78,79,80,81,82,83,84,85,86,87,88,89,90,91,92,93,94,95,96,97,98,99,100,101,102,103,104,105,106,107,108,109,110,111,112,113]. For the twenty-nine studies which compared their GAN model performances with those of other approaches, all of them outperformed the others by 0.1–158.6% [77,78,79,80,81,82,83,84,85,86,87,88,89,91,92,94,95,96,97,98,99,101,103,104,106,107,108,112,113]. The highest accuracy and AUC of GAN-based disease diagnosis were 0.978 [78] and 0.900 [79] for brain MRI-based autism diagnosis, respectively. The performances of GAN-based image reconstruction were as far as 0.991 structural index similarity (SSIM) and 38.36 dB peak signal-to-noise ratio (PSNR) for super-resolution in chest X-ray (CXR) [80], and 31.9 signal-to-noise ratio (SNR) and 21.2 contrast-to-noise ratio (CNR) for abdominal CT denoising [82]. For the top performing GAN-based image segmentation models, 0.929 Dice similarity coefficient (DSC) and 0.338 mm Hausdorff distance (HD) for prostate CT segmentation [86], 0.86 Jaccard index, 0.92 sensitivity and 0.94 PPV for echocardiography segmentation [85], and 0.998 specificity and NPV for cardiac MRI segmentation were achieved [87]. The GAN-based image synthesis and data augmentation for training models of DL-CAD of pneumonia based on CXR boosted the AUC, sensitivity, PPV, F1 score, specificity, and accuracy up to 0.994 [96], 0.993, 0.990, 0.991, [95], 0.97 [92] and 0.990 [94], respectively. The use of GAN for image translation from brain MRI to CT images achieved as far as 0.93 SSIM [110], 0.98 DSC, 32.6 dB PSNR and 42.4 Hounsfield unit (HU) mean absolute error (MAE) [111]. For GAN-based domain adaptation (image translation) in brain MRI segmentation, up to 0.86 DSC, 13.03 mm HD, and 0.23 mm MSD were attained [101]. The application of GAN in automatic image quality assessment yielded 0.81 sensitivity, 0.95 specificity, and 0.93 accuracy [113]. Table 4 summarizes these key findings.

Collectively, the included studies covered pediatric patients aged from 0 to 21 years [77,78,79,80,81,82,83,84,85,86,87,88,89,90,91,92,93,94,95,96,97,98,99,100,101,102,103,104,105,106,107,108,109,110,111,112,113,114]. Their average dataset size for GAN model development was 5799 images (range: 40–17,508 images) [79,80,82,83,89,91,92,94,95,96,98,99,100,101,102,107]/241 scans (range: 5–2187 scans) [77,78,81,84,85,86,87,88,90,93,97,103,104,105,106,108,109,110,111,112,113]. However, no study calculated the required sample size [77,78,79,80,81,82,83,84,85,86,87,88,89,90,91,92,93,94,95,96,97,98,99,100,101,102,103,104,105,106,107,108,109,110,111,112,113]. Except for two studies that collected both public and private datasets [83,97], and one not reporting the dataset source [93], half of the rest (*n* = 17) used public datasets [77,78,79,80,87,89,91,94,95,96,98,99,100,106,107,112,113], and the other half (*n* = 17) collected their own data [81,82,84,85,86,88,90,92,101,102,103,104,105,108,109,110,111]. The most popular public dataset was the chest X-ray dataset consisting of 1741 normal and 4346 pneumonia images of 6087 1–5-year-old children collected from the Guangzhou Women and Children’s Medical Center, China which was used in 10 studies [79,80,89,91,94,95,96,99,100,107].

Nonetheless, about 80% of the included studies (*n* = 29) were retrospective [77,78,79,80,81,82,83,84,87,89,90,91,94,95,96,97,98,99,100,101,102,105,106,107,109,110,111,112,113], and only three were prospective [88,92,103] with the other five not reporting the study design [85,86,93,104,108]. Additionally, about two-thirds of the studies (*n* = 23) did not report the approach for their model internal validation [77,78,79,82,83,85,88,89,90,91,92,93,98,100,101,102,103,104,106,109,110,111,112], and just more than one-fifth (*n* = 8) used the cross-validation to address the small sample size issue [81,84,86,97,99,105,107,108]. Around 90% of studies did not conduct external validation for their models (*n* = 32) [77,78,79,80,81,85,86,88,89,90,91,92,93,94,95,96,97,98,99,100,101,102,104,105,106,107,108,109,110,111,112,113], and compare their model performances with those of clinicians (*n* = 34) [77,78,79,80,81,83,84,85,86,87,88,89,91,92,93,94,95,96,98,99,100,101,102,103,104,105,106,107,108,109,110,111,112,113]. Besides, the reference standard for ground truth establishment was not stated in around half of the included papers (*n* = 17) [77,78,79,83,85,86,89,91,92,93,95,96,100,101,104,107,108].

## 4. Discussion

This article is the first systematic review of the generative AI framework, GAN in pediatric radiology covering MRI [77,78,83,84,87,90,97,101,103,104,105,106,108,109,110,111,112,113], X-ray [79,80,89,91,92,94,95,96,98,99,100,102,107], CT [82,86,93,97], ultrasound [85,88], and PET [81]. Hence, it advances the previous literature reviews about general AI applications [67], and specific uses in radiation dose optimization [26], CAD [29], and chest imaging [28] in pediatric radiology published between 2021 and 2023 which did not focus on the GAN. Unsurprisingly, more than 80% of the studies applied the GAN to MRI and X-ray due to multiplanar imaging capability and excellent soft-tissue contrast of MRI, and less operator dependent and no/low radiation dose for both, resulting in their popularity in pediatric radiology [26,115,116]. Also, it is within expectation that the basic GAN architecture was the most commonly used architecture because it became available earlier than its variants [56,59,63]. The commonest use of basic GAN was for image synthesis and data augmentation [77,78,79,80,81,82,83,84,85,86,87,88,89,90,91,92,93,94,95,96,97,98,99,100,101,102,103,104,105,106,107,108,109,110,111,112,113], which was also one of the most popular GAN applications in the included studies [89,90,91,92,93,94,95,96,97,98,99,100]. These align with the original purpose of the basic GAN which was for the creation of new and original images [63]. CycleGAN was the second most common GAN architecture used in the included studies as the strength of CycleGAN is for image translation without the use of a paired training dataset [62,101,102,109]. A closer look at the findings presented in Table 3 reveals all but two image translation studies used the CycleGAN [101,102,103,104,107,108,109,110,111,112]. It is always challenging to obtain paired datasets to train GAN models for various image translation tasks [102,109]. For example, it is often unrealistic to perform both MRI and CT examinations on the same pediatric patients, resulting in the unavailability of a paired MRI-CT dataset required for training the basic GAN to achieve the image translation from MRI to CT. However, CycleGAN overcomes this issue by using two generators and two discriminators to convert MRI to CT images and vice versa (known as inverse transformation) for creating pseudo image pairs to accomplish the image translation training. In this way, the data collection task becomes easier as only individual MRI and CT images from different patients are required [62,109].

About 80% of the included studies compared their GAN model performances with those of other approaches for benchmarking and indicated that their GAN models outperformed the others [77,78,79,80,81,82,83,84,85,86,87,88,89,91,92,94,95,96,97,98,99,101,103,104,106,107,108,112,113]. Additionally, the absolute performance figures of the best-performing GAN models appear competitive with the other state-of-the-art approaches [77,78,80,82,85,86,87,92,94,95,96,101,110,111,113]. However, the findings from these studies should be used with caution because of the following methodological weaknesses [29]. No study calculated the required sample size for the GAN model development [77,78,79,80,81,82,83,84,85,86,87,88,89,90,91,92,93,94,95,96,97,98,99,100,101,102,103,104,105,106,107,108,109,110,111,112,113]. The sizes of the datasets used were as low as 40 images [83,101]/5 scans [93]. Although the cross-validation internal validation approach can address the small dataset issue to some extent [29], only one-fifth of them used this approach [81,84,86,97,99,105,107,108]. Additionally, just a quarter of the studies covered a wide age range of pediatric patients [84,86,87,93,98,105,109,110,111]. It is well known that there is a lack of generalization ability of many existing DL models because they are only trained by a limited number and variety of patient data [26,50,117]. The variety issue of the included studies was compounded by the retrospective nature of about 80% of them [77,78,79,80,81,82,83,84,87,89,90,91,94,95,96,97,98,99,100,101,102,105,106,107,109,110,111,112,113], and around 60% of these retrospective studies used public datasets which further limited the data variation [77,78,79,80,87,89,91,94,95,96,98,99,100,106,107,112,113]. The most popular public dataset used in the included studies was the one from the Guangzhou Women and Children’s Medical Center, China [79,80,89,91,94,95,96,99,100,107]. However, it is important to note that this public dataset has several image quality issues that could affect the DL model training and hence the eventual performance [118,119]. Hence, the performances of the GAN models covered in this review might not be realized in other settings [26,50,117].

It is noted that no GAN model of the included studies was commercially available [77,78,79,80,81,82,83,84,85,86,87,88,89,90,91,92,93,94,95,96,97,98,99,100,101,102,103,104,105,106,107,108,109,110,111,112,113]. Again, it is within expectation because the GAN has only emerged for 10 years. In contrast, another common DL architecture in medical imaging, convolutional neural network (CNN) which is a deductive AI technique has been available since the 1980s and hence some commercial companies have already used it for developing various products such as Canon Medical Systems Advanced Intelligent Clear-IQ Engine (AiCE) (Tochigi, Japan), General Electric Healthcare TrueFidelity (Chicago, IL, USA), ClariPI ClariCT.AI (Seoul, Republic of Korea), Samsung Electronics Co., Ltd. SimGrid (Suwon-si, Republic of Korea) and Subtle Medical SubtlePET 1.3 (Menlo Park, CA, USA) for radiation dose optimization (denoising) in pediatric CT, X-ray and PET, respectively [1,26].

As a result of the increasing number of GAN publications in pediatric radiology and the popularity of another generative AI application, Chat Generative Pre-Trained Transformer (ChatGPT), it is expected that the GAN would attract the attention of commercial companies to consider using it to develop various applications in pediatric radiology in the future [54,77,78,79,80,81,82,83,84,85,86,87,88,89,90,91,92,93,94,95,96,97,98,99,100,101,102,103,104,105,106,107,108,109,110,111,112,113]. However, based on the previous trend of CNN-based commercial product development for pediatric radiology, such GAN-based commercial solutions should not be available in the coming few years [1,26].

Even when the GAN-based applications are on the market, after several years, developers should disclose their model external validation results, reference standards used for the validation, and their model performances against those of the clinicians on the same tasks for attracting potential customers [29,73,74,120]. According to Table 3, around 90% of the included studies did not conduct external validation for their models [77,78,79,80,81,85,86,88,89,90,91,92,93,94,95,96,97,98,99,100,101,102,104,105,106,107,108,109,110,111,112,113] and compare their model performances with those of clinicians [77,78,79,80,81,83,84,85,86,87,88,89,91,92,93,94,95,96,98,99,100,101,102,103,104,105,106,107,108,109,110,111,112,113]. Besides, the reference standard for ground truth establishment was not stated in around half of the included papers [77,78,79,83,85,86,89,91,92,93,95,96,100,101,104,107,108]. Hence, it would be difficult to earn the pediatric clinicians’ trust in the GAN-based applications for image translation, segmentation, reconstruction, quality assessment, synthesis and data augmentation, and disease diagnosis as there is a lack of trustworthy findings to convince them [77,78,79,80,81,82,83,84,85,86,87,88,89,90,91,92,93,94,95,96,97,98,99,100,101,102,103,104,105,106,107,108,109,110,111,112,113].

There are two major limitations in this systematic review. A single author, despite having experience in performing literature reviews for more than 20 years, selected articles, and extracted and synthesized data [26,29]. As per a recent methodological systematic review, this arrangement is appropriate as the single reviewer is experienced [26,29,70,121,122,123]. Additionally, the potential bias would be addressed to a certain degree due to the use of PRISMA guidelines, data extraction form (Table 3) developed based on the recent systematic reviews on GAN for image classification and segmentation in radiology, and AI for radiation dose optimization and CAD in pediatric radiology, and one narrative review about GAN in adult brain imaging, and QATSDD [26,29,56,62,69,76]. In addition, only English papers were included and this could potentially affect the systematic review comprehensiveness [26,29,72,124,125,126]. Nevertheless, a wider range of applications of GAN in pediatric radiology has been covered in this review when compared with the previous review papers [26,28,29,67].

## 5. Conclusions

This systematic review shows that the GAN can be applied to pediatric MRI, X-ray, CT, ultrasound, and PET for image translation, segmentation, reconstruction, quality assessment, synthesis and data augmentation, and disease diagnosis. About 80% of the included studies compared their GAN model performances with those of other approaches and indicated that their GAN models outperformed the others by 0.1–158.6%. Also, the absolute performance figures of the best-performing models appear competitive with the other state-of-the-art approaches. However, these study findings should be used with caution because of a number of methodological weaknesses including no sample size calculation, small dataset size, narrow data variety, limited use of cross-validation, patient cohort coverage and disclosure of reference standards, retrospective data collection, overreliance on public dataset, lack of model external validation and model performance comparison with pediatric clinicians. More robust methods will be necessary in future GAN studies for addressing the aforementioned methodological issues. Otherwise, trustworthy findings for the commercialization of these models could not be obtained. Additionally, this would affect the clinical adoption of the GAN-based applications in pediatric radiology and the potential advantages of GAN would not be realized widely.

## Figures and Tables

**Figure 1 children-10-01372-f001:**
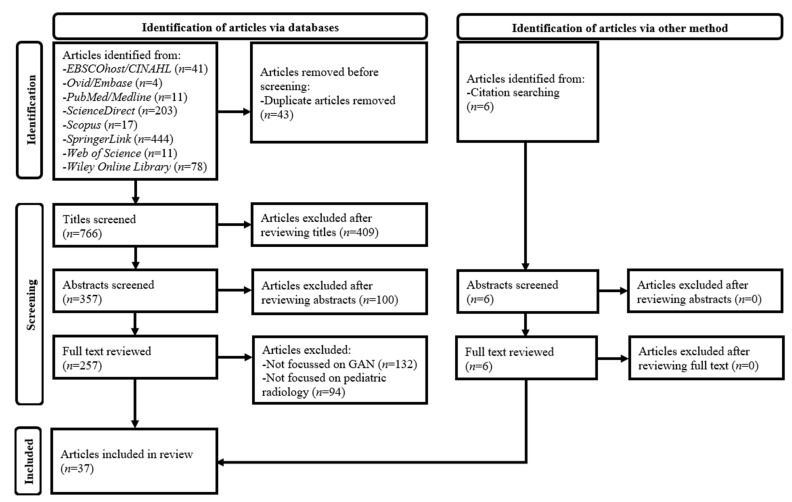
PRISMA flow diagram for the systematic review of the generative adversarial network (GAN) in pediatric radiology.

**Table 1 children-10-01372-t001:** Patient/population, intervention, comparison, and outcome table for the systematic review of the generative adversarial network (GAN) in pediatric radiology.

**Patient/Population**	Pediatric patients aged from 0 to 21 years
**Intervention**	Use of GAN to accomplish tasks involved in pediatric radiology
**Comparison**	GAN versus other approaches to accomplish the same task in pediatric radiology
**Outcome**	Performance of task accomplishment

**Table 2 children-10-01372-t002:** Article inclusion and exclusion criteria.

Inclusion Criteria	Exclusion Criteria
Peer-reviewed original research article Written in English Focused on the use of generative adversarial networks in pediatric radiology	Grey literature Conference abstract Editorial Review Perspective Opinion Commentary Non-peer-reviewed article (e.g., paper on the arXiv platform)

**Table 4 children-10-01372-t004:** Absolute performance figures of best-performing generative adversarial network (GAN) models for various applications in pediatric radiology.

GAN Application	Best Model Performance
Disease diagnosis	0.978 accuracy and 0.900 AUC
Image quality assessment	0.81 sensitivity, 0.95 specificity, and 0.93 accuracy
Image reconstruction	0.991 SSIM, 38.36 dB PSNR, 31.9 SNR and 21.2 CNR
Image segmentation	0.929 DSC, 0.338 mm HD, 0.86 Jaccard index, 0.92 sensitivity, 0.998 specificity and NPV, and 0.94 PPV
Image synthesis and data augmentation for DL-CAD performance enhancement	0.994 AUC, 0.993 sensitivity, 0.990 PPV, 0.991 F1 score, 0.97 specificity, and 0.990 accuracy
Image translation	0.93 SSIM, 0.98 DSC, 32.6 dB PSNR, 42.4 HU MAE, 13.03 mm HD and 0.23 mm MSD

AUC, area under the receiver operating characteristic curve; CAD, computer-aided detection and diagnosis; CNR, contrast-to-noise ratio; DL, deep learning; DSC, Dice similarity coefficient; HD, Hausdorff distance; MAE, mean absolute error; MSD, mean surface distance; NPV, negative predictive value; PPV, positive predictive value; PSNR, peak signal to noise ratio; SNR, signal-to-noise ratio; SSIM, structural index similarity.

## Data Availability

Not applicable.

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
