# Peer review of "Generative Adversarial Network (Generative Artificial Intelligence) in Pediatric Radiology: A Systematic Review"

_children, 2023, doi:10.3390/children10081372_

Round 1

Reviewer 1 Report

The purpose of the author is to systematically review applications 12 of GAN in pediatric radiology, their performances and methods for their performance evaluation.

The article is well written, and the idea is new. The AI is frequently a topic of debate.

I suggest a revision of the article to improve the article comprehension. In particular:

I suggest adding a table that summarize the results;

I suggest reducing the size of table 1;

I suggest reducing the conclusion paragraph in particular eliminating the results

The author could add in the discussion a paragraph about the possible future of GAN in pediatric radiology.

The reference are well organized.

Author Response

1.    The purpose of the author is to systematically review applications of GAN in pediatric radiology, their performances and methods for their performance evaluation. The article is well written, and the idea is new. The AI is frequently a topic of debate.

Response: Thank you for your comment.

2.    I suggest a revision of the article to improve the article comprehension. In particular:
a)    I suggest adding a table that summarize the results;

Response: Thank you for your comment. For addressing this comment, Table 4. Absolute performance figures of best performing generative adversarial network (GAN) models for various applications in pediatric radiology has been added into the Results section for summarizing the findings.

b)    I suggest reducing the size of table 1;

Response: Thank you for your comment. Table 1 (currently Table 3) presents all findings for addressing the review purpose. The suggested size reduction might affect the comprehensiveness of the review. However, I can move this table to the Appendix section if necessary because Table 4. Absolute performance figures of best performing generative adversarial network (GAN) models for various applications in pediatric radiology has been added into the Results section. I hope you will find my response satisfactory. 

c)    I suggest reducing the conclusion paragraph in particular eliminating the results

Response: Thank you for your comment. The following results, “(image translation: 0.93 SSIM, 0.98 DSC, 32.6 dB PSNR, 42.4 HU MAE, 13.03 mm HD and 0.23 mm MSD; segmentation: 0.929 DSC, 0.338 mm HD, 0.86 Jaccard index, 0.92 sensitivity, 0.998 specificity and NPV, and 0.94 PPV; reconstruction: 0.991 SSIM, 38.36 dB PSNR, 31.9 SNR and 21.2 CNR; quality assessment: 0.81 sensitivity, 0.95 specificity, and 0.93 accuracy; synthesis and data augmentation for DL-CAD performance en-hancement: 0.994 AUC, 0.993 sensitivity, 0.990 PPV, 0.991 F1 score, 0.97 specificity and 0.990 accuracy; and disease diagnosis: 0.978 accuracy and 0.900 AUC)” have been removed from the Conclusion section for addressing this comment.

d)    The author could add in the discussion a paragraph about the possible future of GAN in pediatric radiology.

Response: Thank you for your comment. The following paragraph has been added into the Discussion section for addressing this comment.

“As a result of the increasing number of GAN publications in pediatric radiology and the popularity of another generative AI application, Chat Generative Pre-Trained Transformer (ChatGPT), it is expected that the GAN would attract the attention of commercial companies to consider using it to develop various applications in pediatric radiology in the future [54,77-113]. However, based on the previous trend of CNN based commercial product development for pediatric radiology, such GAN based commercial solutions should not be available in the coming few years [1,26].”

3.    The reference are well organized.

Response: Thank you for your comment.

Reviewer 2 Report

Thank you for taking the time to create this interesting article. I present my review below in bullet points. Please refer to all points and if possible modify the article as suggested.

1. In my opinion, it is worth mentioning a few more interesting articles on the application of AI in radiology. A paragraph on the impact of the COVID-19 pandemic on the increased interest in AI in radiology should be added, in particular on AI related to classification or detection on medical imaging data. Please add the following articles treating this in this paragraph:

10.1016/j.radi.2020.09.010

10.1016/j.cmpbup.2021.100025

10.1002/jmv.28787

10.7150/ijms.76515

doi.org/10.3390/healthcare11091204

In addition, an important aspect has been overlooked - AI-supported mobile solutions in radiology. This topic has been very hot recently.

10.1016/j.compbiomed.2022.105298

10.3390/healthcare10102040

10.1371/journal.pone.0221720

Please also add a citation to this extremely important article:

10.3390/healthcare10010154

2. The methodology section lacks information on the inclusion and exclusion criteria for articles. This section should be clearly separated from the rest of 2.2 Article Selection and provide the inclusion and exclusion criteria for the study.

3. After only a cursory check using the given keywords in section 2.1 in PubMed, the results are different from those given in Figure 1. Have you perhaps applied filters? If so, please specify them for each of the databases searched.

4. If it is a systematic review has it been registered in PROSPERO or another database? Are there similar systematic reviews in this database?

5. In the results section, it is unnecessary to list the quantitative share of articles from a given country. This is all visible in the table and creates unnecessary chaos. Please remove this section.

6. Why is the PICO scheme table not shown anywhere?

7. Has a protocol for this study been established?

8. Do the authors have plans to conduct a meta-analysis of the data presented?

Please quote all the articles I have selected. Making changes. I wish you good luck in your further scientific work.
With kind regards. Reviewer.

Author Response

Thank you for taking the time to create this interesting article. I present my review below in bullet points. Please refer to all points and if possible modify the article as suggested.

Response: Thank you for your comment.

1.    In my opinion, it is worth mentioning a few more interesting articles on the application of AI in radiology. A paragraph on the impact of the COVID-19 pandemic on the increased interest in AI in radiology should be added, in particular on AI related to classification or detection on medical imaging data. 

Please add the following articles treating this in this paragraph:

10.1016/j.radi.2020.09.010
10.1016/j.cmpbup.2021.100025
10.1002/jmv.28787
10.7150/ijms.76515
doi.org/10.3390/healthcare11091204
In addition, an important aspect has been overlooked - AI-supported mobile solutions in radiology. This topic has been very hot recently.

10.1016/j.compbiomed.2022.105298
10.3390/healthcare10102040
10.1371/journal.pone.0221720

Please also add a citation to this extremely important article:

10.3390/healthcare10010154

Response: Thank you for your comment. The first sentence of the Introduction section has been changed to “Artificial intelligence (AI) is a popular topic in radiology such as for rapid disease (e.g., COVID-19) detection on various platforms including mobile devices [1-12]” to cover these important areas with the 9 suggested references cited for addressing this comment.

2.    The methodology section lacks information on the inclusion and exclusion criteria for articles. This section should be clearly separated from the rest of 2.2 Article Selection and provide the inclusion and exclusion criteria for the study.

Response: Thank you for your comment. Table 2. Article inclusion and exclusion criteria has been added into the Materials and Methods section for addressing this comment.

3.    After only a cursory check using the given keywords in section 2.1 in PubMed, the results are different from those given in Figure 1. Have you perhaps applied filters? If so, please specify them for each of the databases searched. 

Response: Thank you for your comment. I used the PubMed database to carry out another article search with the keywords stated in Section 2.1 and without any filter on 4th August 2023. The number of articles identified was 13 which is different from the number stated in Figure 1, 11. However, the 2 additional articles were published after my previous literature search (https://doi.org/10.1148/radiol.230052 published on 5th July 2023 and https://doi.org/10.1186/s41747-023-00343-y published on 7th June 2023). Both did not focus on pediatric radiology. Please let me know if you need a copy of the PudMed search result. I hope you will find my response satisfactory. 

4.    If it is a systematic review has it been registered in PROSPERO or another database? Are there similar systematic reviews in this database?

Response: Thank you for your comment. As per the Instructions for Authors of the Children journal (https://www.mdpi.com/journal/children/instructions), systematic review registration is not required. Hence, this systematic review is not registered. Also, I used the PROSPERO system to carry out the registration search with the keyword, “generative adversarial network”. Four records were found as follows.

1. A systematic review of Deep Learning methodologies used in the drug discovery process with emphasis to the in vivo validation [CRD42022329949]
2. GAN for the diagnosis of Alzheimer's disease: a systematic review and meta-analysis [CRD42021275294]
3. Generative adversarial networks for the diagnosis of Alzheimer's disease: a systematic review [CRD42021274331]
4. Inter-modality Image-to-image translation of CT and MRI brain images – a systematic review. [CRD42022368642]

Hence, there is no similar systematic review in the PROSPERO database. Also, the status of all above reviews is ongoing. I hope you will find my response satisfactory.

5.    In the results section, it is unnecessary to list the quantitative share of articles from a given country. This is all visible in the table and creates unnecessary chaos. Please remove this section.

Response: Thank you for your comment. The following sentence of the Results section, “The authors of the papers were from USA (n=16) [77,78,84,86-88,92,94,97-104], China (n=9) [72,76,82,83,89,92,97,98,103], Canada (n=5) [70,71,75,79,88], India (n=4) [68,69,80,91], Germany (n=3) [81,93,94], Australia (n=2) [68,69], France (n=2) [74,75], The Netherlands (n=2) [95,96], United Arab Emirates (n=2) [68,69], Austria (n=1) [93], Croatia (n=1) [93], Egypt (n=1) [85], Hungary (n=1) [90], Japan (n=1) [97], Republic of Korea (n=1) [73] and Romania (n=1) [90].” has been removed for addressing this comment.

6.    Why is the PICO scheme table not shown anywhere?

Response: Thank you for your comment. Table 1. Patient/population, intervention, comparison, and outcome table for systematic review of generative adversarial network (GAN) in pediatric radiology has been added into the Materials and Methods section for addressing this comment.

7.    Has a protocol for this study been established?

Response: Thank you for your comment. The systematic review protocol is incorporated into the Materials and Methods section rather than presenting it separately. This is because the systematic review registration is not required by the Children journal (https://www.mdpi.com/journal/children/instructions) and hence unnecessary to have a separate protocol. I hope you will find my response satisfactory. 

8.    Do the authors have plans to conduct a meta-analysis of the data presented?

Response: Thank you for your comment. As stated in the Materials and Methods section, “Meta-analysis was not performed as this systematic review included a range of GAN applications, resulting in high study heterogeneity which would affect its usefulness”. However, it appears a good idea to conduct a meta-analysis when more original articles on a specific application in pediatric radiology have become available in the future. I hope you will find my response satisfactory. 

Please quote all the articles I have selected. Making changes. I wish you good luck in your further scientific work.
With kind regards. Reviewer.

Response: Thank you for your comment. All 9 suggested articles have been cited and changes have been made for addressing this comment.

Round 2

Reviewer 2 Report

All questions were answered by the authors. Any suggested amendments have been made. The text has been checked for any changes. I recommend the article for publication.